Semantic-visual shared knowledge graph for zero-shot learning

Yu Beibei
Xie Cheng xiecheng@ynu.edu.cn
Tang Peng
Li Bin
School of Software, Yunnan University , Kunming , Yunnan , Chain
Ashraf Imran
Electronic publication date: 2023 Mar 22
Publication date: 2023
Volume: 9
Electronic Location ID: e1260
Received 2022 Jul 21; Accepted 2023 Jan 31
Copyright: ©2023 Yu et al.
Copyright year: 2023
Copyright holder: Yu et al.
License: This is an open access article distributed under the terms of the Creative Commons Attribution License, which permits unrestricted use, distribution, reproduction and adaptation in any medium and for any purpose provided that it is properly attributed. For attribution, the original author(s), title, publication source (PeerJ Computer Science) and either DOI or URL of the article must be cited.
License URL: https://creativecommons.org/licenses/by/4.0/

Keywords: Zero-shot learning, Knowledge graph, Multi-modal learning, Image classification

Funding: The National Natural Science Foundation of China 62106216 62162064 The work is supported by the National Natural Science Foundation of China (No. 62106216 and No. 62162064). The funders had no role in study design, data collection and analysis, decision to publish, or preparation of the manuscript.

==============================
Almost all existing zero-shot learning methods work only on benchmark datasets (e.g., CUB, SUN, AwA, FLO and aPY) which have already provided pre-defined attributes for all the classes. These methods thus are hard to apply on real-world datasets (like ImageNet) since there are no such pre-defined attributes in the data environment. The latest works have explored to use semantic-rich knowledge graphs (such as WordNet) to substitute pre-defined attributes. However, these methods encounter a serious “role=“presentation”>domain shift” problem because such a knowledge graph cannot provide detailed enough semantics to describe fine-grained information. To this end, we propose a semantic-visual shared knowledge graph (SVKG) to enhance the detailed information for zero-shot learning. SVKG represents high-level information by using semantic embedding but describes fine-grained information by using visual features. These visual features can be directly extracted from real-world images to substitute pre-defined attributes. A multi-modals graph convolution network is also proposed to transfer SVKG into graph representations that can be used for downstream zero-shot learning tasks. Experimental results on the real-world datasets without pre-defined attributes demonstrate the effectiveness of our method and show the benefits of the proposed. Our method obtains a +2.8%, +0.5%, and +0.2% increase compared with the state-of-the-art in 2-hops, 3-hops, and all divisions relatively.

Introduction

In recent years, zero-shot learning has attracted widespread attention in computer vision and machine learning areas. It aims to predict new classes that appear during the training process. The base idea of zero-shot learning is to use labeled semantic information (normally word attributes) to learn a projection between semantic space and visual space. Then, visual samples from new classes can be projected into semantic space to match the attributes to decide the corresponding classifications. In the past five years, a large number of methods for zero-shot learning have been proposed based on this idea  (Liu et al., 2022; Kim, Lee & Byun, 2021; Chen et al., 2021; Chen et al., 2020b; Annadani & Biswas, 2018; Li et al., 2020; Yu & Lee, 2019; Liu et al., 2019; Cacheux, Borgne & Crucianu, 2019).

However, these zero-shot learning methods inevitably need to follow a major premise that semantic attributes should be pre-defined and labeled in the datasets. Almost all recent zero-shot learning works are only evaluated on six small benchmark datasets (CUB, SUN, aPY, FLO, AwA1 and AwA2). Table 1 provides the statistics of these datasets. It can be observed the semantic attributes are pre-defined in each dataset from 64 (aPY) to 1024 (CUB) which describe fine-grained semantics for all classes like “Wings-Color”, “Leg-color”, “Breast”, etc. Moreover, these datasets are quite small, only tens or hundreds classes and tens of thousands samples, which are far away from real-world data environment, normally tens of thousands classes and millions samples. Thus, existing zero-shot methods are hard to be applied in real-world data environments such as ImageNet which does not provide any pre-defined attributes for any classes.

Table 1 Six traditional benchmark datasets with pre-defined attributes, for zero-shot learning only.

	Seen	Seen	Unseen	Unseen	Attributes	
	classes	samples	classes	samples		
aPY	20	15,399	12	7,924	64	
AwA1	40	30,475	10	5,685	85	
AwA2	40	37,322	10	7,913	85	
FLO	82	8,189	20	1,155	1,024	
CUB	150	11,788	50	2,967	1,024	
SUN	645	14,340	72	1,440	102	
Comparing with real-world dataset	
ImageNet	1K	1.29M	20K	12.9M	0	

In the past four years, the graph neural network (GNN) has been adopted by zero-shot learning, which makes zero-shot learning tasks applicable in real-world data environments (Wang, Ye & Gupta, 2018a; Kampffmeyer et al., 2019a; Wang & Jiang, 2021a). This is because GNN-based methods use open knowledge (Wikipedia, freebase, Nell, etc.) to substitute pre-defined attributes in semantics and have better generalization power in semantic representation. However, these GCN-based zero-shot learning methods for image classification on the ImageNet dataset, like GCNZ (Wang, Ye & Gupta, 2018a), DGP (Kampffmeyer et al., 2019a) and CL-DKG (Wang & Jiang, 2021a), still have a long way from the traditional methods evaluated in the six small benchmark datasets, which normally achieve more than 50% Top-1 accuracy. While the out-of-state method (Wang & Jiang, 2021a) only obtains 1.0% Top-1 accuracy and 16.7% Top-20 accuracy on ImageNet dataset. The main finding of our analysis is that, in comparison to the pre-defined attributes in the six benchmark datasets, the knowledge network used in GCN-based approaches offers significantly less fine-grained semantics. This leads a serious “domain-shift” problem (Min et al., 2020; Fu et al., 2015; Zhu et al., 2018; Wan et al., 2019; Ni, Zhang & Xie, 2019) during projection between semantic space and visual space. In other words, an existing knowledge graph can only provide coarse-grained semantics (such as name, habitat, taxonomy, hypernym, hyponym, etc.) without fine-grained semantics (like head, tail, beak, legs, etc.). Thus, the core question becomes “can a knowledge graph provide fine-grained semantics or even more for zero-shot learning?”.

Based on the above discussion, we are going to present a solution that lets fine-grained semantics be extracted from raw images and shared with the existing knowledge graph. The idea is that there are lots of fine-grained semantics already hidden in the image samples, especially in the large-scale image set. As the example shown in Fig. 1, when a person sees an image sample about “Indigo Bunting”, he can quickly establish fine-grained semantics about this bird in his brain, such as “has deep and light blue head”, “has black and light blue tail”, “has deep blue breast”. Then, a large fine-grained semantic network (graph) can be further established if there are enough image samples have been seen. On the contrary, these fine-grained semantics can also complement semantic representations.

Figure 1 Fine-grained semantics can be extracted from a raw image sample.

It can be further used to establish a fine-grained knowledge graph through extracting the amount of image samples.

Based on the idea, we first scan all the image samples of seen classes in ImageNet-1K to extract parts of visual features for each seen class. Then, WordNet nodes and relations are extracted according to the seen and unseen class labels in ImageNet-1K and embedded as semantic features by a word embedding model. After that, parts of visual features and WordNet semantic features are connected together as a semantic-visual shared knowledge graph (SVKG). At last, a multi-modals GCN network is proposed to embed (SVKG) into graph representations that can be used in zero-shot learning tasks, as showed in Fig. 2. Experimental results on real-world datasets without pre-defined attributes demonstrate the effectiveness of our method and show the benefits of the proposed semantic-visual shared knowledge graph.

Figure 2 The overview of the proposed method.

It can be divided into three modules. The first module, semantic-visual shared knowledge graph, is introduced in section “Semantic-visual Shared Knowledge Graph”; The second module, multi-modal GCN network, is detailed in section “Multi-modals Graph Representation Learning”; The last module, zero-shot learning, is explained in section “Cross-modals Zero-shot Learning”.

Overall, the main contributions of this paper are summarized as follows:

• We first propose the semantic-visual shared knowledge graph that stores high-level information in semantic features while storing fine-grained information in parts visual features. It can be used for real-world zero-shot learning tasks without requiring pre-defined attributes.

• We propose a multi-modal GCN network to fuse semantic modal and visual modal together in the same knowledge graph. The output of the network is semantic-visual shared graph representation that can be easily used for downstream zero-shot learning tasks.

Related Work

Zero-shot learning based on attributes

Attribute-based methods account for the largest proportion of zero-shot learning (ZSL) research since ZSL was first proposed in DAP (Lampert, Nickisch & Harmeling, 2009) in 2009 with a proposed dataset called AwA which contains pre-defined attributes. It principally annotates the attributes of images (such as whether there is a tail, hair color, etc.), then learn the semantic attribute characteristics of visual objects, and finally judge whether the attribute combinations are satisfied by the visual objects.

Following the design principles of AwA dataset, a series of benchmark datasets with pre-defined attributes are established today, including AwA2, CUB, SUN, FLO and aPY. Based on these benchmark datasets, ZSL has a blowout development with several milestones.

In 2013, with the development of semantic embedding technology, the first milestone of ZSL was the ALE model proposed by Akata et al. (2013) which can encode the attributes as semantic vectors, encode images as visual vectors, and then learn a function to calculate the similarity between semantic vectors and visual vectors, so as to match the corresponding mappings between attributes and images.

In 2017, with the development of deep learning technology in the field of visual computing, the second milestone of ZSL is the SAE model proposed by Kodirov, Xiang & Gong (2017). By using the Auto-Encoder network, it can encode more fine-grained attribute features and visual features, so as to better achieve “semantic-to-visual” matching. The overall performance of SAE is significantly improved compared with ALE.

In 2018, with the remarkable performance of Generative Adversarial Networks (GAN) (Goodfellow et al., 2020) in image processing, ZSL achieved the third milestone with the representative model GAZSL proposed by Zhu et al. (2018). It uses GAN to synthesize fake visual features from semantic features, and then match the fake visual features with the real visual features to predict the unseen objects.

Recently, some researchers have started to investigate how to mix various types of semantics (attributes, word2vec, text description, knowledge graph) to address further “domain shift” issues. For instance, Wang et al. (2021) learns discriminative classifiers using a variety of semantic viewpoints. In Xie et al. (2021) and Naeem et al. (2021), open knowledge is regarded as auxiliary or augmented semantics added with pre-defined features.

However, the above methods use pre-defined properties as their primary semantic source. As a result, these methods are essentially restricted to benchmark datasets with pre-defined features. The applicability of these methods in real-world environments devoid of pre-defined features is restricted to a small number of cases.

Zero-shot learning based on knowledge graph

Knowledge graph (KG) actually is a third part knowledge base that can provide semantic information for semantic-to-visual transformation in zero-shot learning. Knowledge graph (KG) actually constitutes a third-party knowledge base that can provide semantic information for semantic-to-visual transformation in zero-shot learning. Thus, KG is intuitively considered a substitution for pre-defined attributes. Benefiting from the development of GNN, GCNZ (Wang, Ye & Gupta, 2018a) creates a formal knowledge graph-based on WordNet to substitute pre-defined attributes and learn semantic embedding from structure information and word embedding. It is, indeed, the first work that tries to apply knowledge graphs and GNN as the backbone for the real-world dataset zero-shot learning task. Based on Wang, Ye & Gupta (2018a) and Kampffmeyer et al. (2019a) proposes a so-called Dense Graph Propagation (DGP) method to gather the semantic information through the relations of the knowledge graph. CL-DKG (Wang & Jiang, 2021a) applies contrastive learning on dual knowledge graph to learn the projection between semantic space and visual space without any pre-defined attributes. It exploits multiple knowledge relationships among classes simultaneously to learn robust and discriminative classifiers for unseen classes.

However, compared to the pre-defined features in the six benchmark datasets, the knowledge network used in GNN-based approaches (Wang, Ye & Gupta, 2018a; Kampffmeyer et al., 2019a; Liu et al., 2020; Wang et al., 2021) provides far less fine-grained semantics. In contrast to them, we propose a solution that enables fine-grained semantics to be extracted from raw images and shared with the existing knowledge graph in order to enhance recognition performance in the ZSL challenge.

Visual knowledge applied for zero-shot learning

To enhance the fine-grained semantic information for existing KG, the latest research tries to apply “visual knowledge” in zero-shot learning. Zhu, Wang & Saligrama (2019) propose a novel low-dimensional embedding for visual objects called “visually semantic” to narrow the semantic gap between high-dimensional visual space and semantic space in zero-shot learning. Xu et al. (2020) proposed a zero-shot learning framework that jointly learns discriminative visual semantics only using class-level attributes. Liu et al. (2021) propose a goal-oriented gaze estimation module (GEM) to improve the discriminative attribute localization based on the class-level attributes for ZSL. They introduce so-called “human gaze locations” to obtain new regions of visual semantics. Xie et al. (2020) extracts the relationships among visual regions to enhance the semantic information for both seen and unseen classes and then can lunch region-based relational reasoning for ZSL. Song & Zhang (2022) consists of two graphs constructed by semantic and visual representations respectively to enhance semantic representations.

However, the above methods continue to see visual semantics as auxiliary or augmented semantics. In contrast to them, we want to combine semantic and visual representations into a single common knowledge graph, so-called visual-semantic shared knowledge graph.

Methodology

Problem definition

Knowledge graph-based Zero-Shot learning uses the knowledge-aided method to learn an image classification model from seen classes to predict unseen classes. First, given a knowledge graph G=V,E , Then, we have X ∈ ℝN×F be the word-embedding set for all the nodes in V, and A ∈ ℝN×N be the adjacency matrix transferred from E. Here, F is the dimensions of the embedding. After, a graph representation model gΘ(X, A, Iseen) is learned to transfer X to the graph node representation H ∈ ℝN×F′ supervised by Iseen. Here, Iseen ∈ ℝN×F′′ is the seen image feature set extracted by a CNN model. At last, an image classification model LΘH,Iunseen is learned to classify Iunseen to the particular class according to the graph node representation H. Here, Iunseen ∈ ℝN×F′′ is the unseen image feature set extracted by a CNN model.

Semantic-visual shared knowledge graph

Semantic-visual shared knowledge graph (SVKG) is a multi-modal graph that contains both semantic embedding and visual embedding in the same graph. Let X represents the embedding set of the graph nodes with Xs denotes Word-Embedding nodes and Xv represents CNN-Embedding nodes. A represents the edge set while As denotes the edges among Word-Embedding nodes and Av denotes the edges among CNN-Embedding nodes. SVKG is defined in Algorithm 1.

________________________________________________________________________________ Algorithm 1 Semantic-visual shared knowledge graph ________________________________________________________________________________ Input: G = (V,E), seen images Output: SV KG  1:  Xs ←− Put X into Glove  2:  As ←− Put X into WordNet hyponym/hypernym  3:  Xv ←− Put seen images into EfficientDet  4:  As ←− Put Xv connecte with semantic object node  5:  X′ ←− Xs ∪ Xv  6:  A′ ←− As ∪ Av  7:  SV KG ←−{ X′, A′ }  8:  return  SV KG  _______________________________________________________________________________

In SVKG, the semantic node set Xs is constructed from WordNet (https://wordnet.princeton.edu/) Noun words and embedded into word features by Glove (https://nlp.stanford.edu/projects/glove/). The edge matrix As is established by WordNet hyponym/hypernym links among these words. The visual node set Xv is obtained by using EfficientDet (Tan, Pang & Le, 2020) to detect fine-grained parts visual features (such as head, back, belly, breast, leg, etc.) from ImageNet-1k. Then, these visual nodes are connected to their semantic object node by the edge matrix Av (such as “hasHead”, “hasBelly”, etc.). An example of SVKG about bird Finch is presented in Fig. 3.

Figure 3 An example of a semantic-visual shared knowledge graph (SVKG) about the finch.

Graph augmentation for SVKG

In the zero-shot learning process, semantic feature space Xs needs to be transferred into semantic-visual shared space H. However, the visual feature space (fine-grained visual parts) Xv might cause interference during Xs⟶H transfer process, leading to a serious over-fitting problem. Empirically, many data augmentation methods have shown to improve the generalization and robustness of the learning model (Zhao et al., 2021; Rong et al., 2020; Srivastava et al., 2014; Chen et al., 2020a). Thus, in this work, we introduce an extra graph node with zero-initialized features to augment SVKG. This zero-initialized node Xaug then links to all seen nodes by the edge matrix Aaug. After that, the node is indirectly connected to all visual feature nodes Xv through the corresponding seen nodes. It can moderate the strength of feature propagation from visual feature node Xv to alleviate the over-fitting problem. Augmentation graph SVKGaug is defined in Eq. (1). (1) SVKGaug=X′,A′X′=X∪Xaug,A′=A∪Aaug

where Xaug is the augmentation node with zero-initialized feature. Aaug is the adjacency matrix to link Xaug to all seen nodes. And X′, A′ represents feature matrix and adjacency matrix of SVKG, respectively. An example of SVKGaug is presented in Fig. 4.

Figure 4 An example of augmentation knowledge graph (SVKGaug) about the finch.

Multi-modals graph representation learning

As explained in the ‘Problem Definition’ section, knowledge graph SVKGaug needs to be represented in a particular feature space to support the downstream task (zero-shot image classification). In SVKGaug, there are two feature modals. One is semantic feature modal {Xs generated from Glove-Embedding. The other is visual feature modal Xv extracted by EfficientDet. Thus, multi-modals graph representation learning is aimed to transfer both Xs and Xv into a semantic-visual shared feature modal H.

More specifically, we propose a dual ways propagation graph convolution network to transfer Xs and Xv into H, as defined in Eq. (2). X⟶H⟶H′

(2) H=gΘσDs−1As∪AaugX

H′=gΘσDv−1Av∪AaugH

where gΘ is a Graph Convolutional Network (GCN) employed from Kipf & Welling (2017). σ⋅ represents a nonlinear activation function. D is the diagonal degree matrix of A matrix, where D(i,i) = ∑jA(i,j). H′ is the semantic-visual shared graph representation feature set that can be used for downstream task.

Cross-modals zero-shot learning

So far, we already obtained the final representation H′ for the knowledge graph SVKGaug. Each Hi′ represents the centroid feature of the corresponding class (this class could be seen class or unseen class). Here, H′ is knowledge graph feature modal. However, in the zero-shot image classification, the targets are the real images which is an image feature modal. Thus, the problem becomes a cross-modal transfer problem (Zhuang et al., 2021). Let I represents the real image features. Iseen and Iunseen denote seen classes features and unseen classes features relatively. Ii,seen represents the centroids feature of ith seen class. Then, we have following the loss function, Eq. (3), to force knowledge graph features H′ close to real image features I. (3) H′⟶differences⟵IL=1N∑i=1NHi′−Ii,seen2.

The base idea is simple. In the training process, the graph features H′ could be updated supervised by real image features I. We calculate the similarities for all (Hi′−Ii,seen) pairs and use average similarity differences as the loss function to train the zero-shot classification model. Note that, in the experiment, a pre-trained Resnet-50 model is used to extract real image features I.

In the predicting process, we first use the same Resnet-50 model to extract unseen image feature Iunseen (Algorithm 2 Input). Then, Iunseen is input into the trained classification model to calculate the similarities with all graph features H′. To obtain the predicting result, Resultset records all the similarities between H′ and Iunseen, and the labeli of corresponding Hi′ (Algorithm 2 2-6). After, the top-k highest similarities with labels are selected as the zero-shot classification result (Algorithm 2 7-9). Algorithm 2 presents the whole zero-shot predicting process.

________________________________________________________________________________________ Algorithm 2 Zero-shot Predicting Process _______________________________________________________________________________________ Input: H′, Iunseen Output: Top-k result  1: Resultset ←−{∅, ∅}  2: for H′i in H′ do  3:   Simi ←− H′i ⋅ Iunseen  4:   lableli ←− findLabel(H′i)  5:   Resultset ←−{Simi, lableli}  6: end for  7: Resultset ←− SortBySim(Resultset)  8: Resultset ←− Top-k(Resultset)  9: return  Resultset  ____________________________________________________________________________________

It can be observed, the unseen image feature Iunseen is not used during the whole training process. In the predicting process, the input Iunseen is the first time the model touches the unseen images.

Experiments

In this section, the datasets and evaluation metrics are introduced first. Then, the implementation details about the model settings are explained. After, the state-of-the-arts comparisons are presented. Then, an ablation study is conducted for the proposed SVKG. At last, extra downstream tasks and observations are discussed.

Datasets and evaluation metrics

The experiments are conducted on ImageNet (Deng et al., 2009), which is a real-world dataset and also the most large-scale benchmark for zero-shot image classification. The divisions for zero-shot tasks are 1K seen classes from ImageNet-2012-1K while 1.5K, 7.8K and 20K unseen classes for 2-hops, 3-hops and all from ImageNet. These divisions are grouped together in our experiments named “General” group. Here, n-hops means the most nth jumps to connect to other classes of ImageNet through the relations of a WordNet graph. For example, as showed in Fig. 2, given a seen class “Indigo Bunting”, 2-hops can be unseen classes “Ortolan” and “Finch”, but 3-hops will has extra unseen classes including “Chaffinch”, “Redpoll” and “Grosbeak”. Note that there is no overlap between the seen and unseen classes in the three divisions.

In order to show the effectiveness of visual feature Xv in SVKG, a “Detailed” group which contains four subsets (‘bird’, ‘snake’, ‘primate’ and ‘dog’ categories) are divided from ImageNet-1K and corresponding parts visual features Xv for all these detailed categories are extracted from their image samples by EfficientDet. In a short, the experiments involve two groups and seven divisions divided by ImageNet. The detailed information for each corresponding division is shown in Table 2.

Table 2 The datasets divided from ImageNet.

Group	Division	Train	Test	label	
		cls	samples	cls	samples		
General	2-hops	1k	1.29M	1.5k	1.3M	38.6%	
3-hops	1k	1.29M	7.8k	5.8M	11.3%	
All	1k	1.29M	20k	12.9M	4.6%	
Detailed	bird	58	74k	635	579k	8.4%	
snake	17	17k	98	76k	14.8%	
primate	19	25k	51	35k	27.1%	
dog	118	155k	77	73k	60.5%	
Notes.

The “Detailed” group can be found in the Supplemental File.

There, the label rate demonstrates the difficulty of the corresponding division. This is because with the reduction of supervision in semi-supervised manner, there are higher requirements for the generalization and robustness of the model. The performance of corresponding division will also degrade as the supervised label rate decreases. Therefore, the accuracy in widely different divisions can reflect the robustness of the model. As can be seen from Table 2, the most difficult task is the all division. The label rate of the four subsets increases in turn. Among them, the bird division simulates the division of “All”, which most intuitively illustrates the impact of fine-grained semantics. In contrast, the dog division is completely different from the “All” division where the majority of classes are seen.

The proposed method is evaluated according to Generalized Zero-Shot Learning (GZSL) setting, which is the most challenging evaluation metric for zero-shot learning. In GZSL, all classes, no matter seen or unseen classes, are all considered as the candidate classes in the testing, while the test samples are only from unseen classes. We adopt the same train/test splits and the Top-k Hit Ratio (Hit@k) metric, in accordance with Wang, Ye & Gupta (2018b), Kampffmeyer et al. (2019b) and Wang & Jiang (2021b) for all divisions.

Implementation details

In the model implementations, Resnet-50 (He et al., 2016) is used as feature extractor to extract 2048-dimensions visual feature that has been pre-trained on the ImageNet 2012 dataset. GloVe (Pennington, Socher & Manning, 2014) is used to extract 300-dimensions semantic embedding for initial graph representation on WordNet nodes. EfficientDet (Tan, Pang & Le, 2020) is used to extract 300-dimensions visual features for initial graph representation on fine-grained part nodes. The proposed method is trained for 1000 epochs using ADAM Kingma & Ba (2015) optimizer with learning rate 0.001 and weight decay 0.0005. For each convolutional layer, we employ Dropout operation and leaky ReLUs with a dropout rate of 0.5 and a negative slope of 0.2 respectively. Besides, the model is implemented by PyTorch, training on 4 ×GTX-2080Ti GPUs.

Comparisons with state-of-the-art

In this section, all the comparisons conducted in both General and Detailed groups follow the GZSL setting. For General group, DeViSE (Frome et al., 2013), ConSE (Norouzi et al., 2014), ConSE2 (Wang, Ye & Gupta, 2018b), GCNZ (Wang, Ye & Gupta, 2018b), DGP (Kampffmeyer et al., 2019b) and CL-DKG (Wang & Jiang, 2021b) are selected as counterparts. The results of these counterparts are directly copied from their original publications. For Detailed group, we apply the sources code of each counterpart to conduct the comparison in four sub-categories datasets. Only DGP, GCNZ and SGCN (Kampffmeyer et al., 2019b) are compared since other methods have not provided the source code.

General group comparisons: From Table 3, it can be observed that our method achieves a new state-of-the-art on Top-1 accuracy in all divisions of ImageNet. Especially, our method obtains +2.8%, +0.5% and +0.2% increasing compared with DGP in 2-hops, 3-hops and all divisions relatively. It is a significant progress since Top-1 accuracy in ImageNet zero-shot learning is the most challenging and representative task. The result also demonstrates that fine-grained semantic and visual features guide model generates more discriminative graph representations, which are helpful for recognizing fine-grained unseen classes.

Table 3 Top-k evaluation in general group.

Test set	Model	hit@k(%)	
		1	2	5	10	20	
2-hops(+1k)	DeViSE (Frome et al., 2013)	0.8	2.7	7.9	14.2	22.7	
ConSE (Norouzi et al., 2014)	0.3	6.2	17	24.9	33.5	
ConSE2 (Wang, Ye & Gupta, 2018b)	0.1	11.2	24.3	29.1	32.7	
GCNZ (Wang, Ye & Gupta, 2018b)	9.7	20.4	42.6	57	68.2	
DGP (Kampffmeyer et al., 2019b)	10.3	26.4	50.3	65.2	76	
CLDGK (Wang & Jiang, 2021b)	7.0	26.8	52.5	67.5	77.9	
ours	13.1	25.7	47.0	61.0	72.4	
3-hops(+1k)	DeViSE (Frome et al., 2013)	0.5	1.4	3.4	5.9	9.7	
ConSE (Norouzi et al., 2014)	0.2	2.2	5.9	9.7	14.3	
ConSE (Wang, Ye & Gupta, 2018b)	0.2	3.2	7.3	10	12.2	
GCNZ (Wang, Ye & Gupta, 2018b)	2.2	5.1	11.9	18	25.6	
DGP (Kampffmeyer et al., 2019b)	2.9	7.1	16.1	24.9	35.1	
CLDGK (Wang & Jiang, 2021b)	2.0	7.1	17.3	26.2	36.5	
ours	3.4	6.9	14.9	22.7	32.2	
All(+1k)	DeViSE (Frome et al., 2013)	0.3	0.8	1.9	3.2	5.3	
ConSE (Norouzi et al., 2014)	0.2	1.2	3	5	7.5	
ConSE (Wang, Ye & Gupta, 2018b)	0.1	1.5	3.5	4.9	6.2	
GCNZ (Wang, Ye & Gupta, 2018b)	1.0	2.3	5.3	8.1	11.7	
DGP (Kampffmeyer et al., 2019b)	1.4	3.4	7.9	12.6	18.7	
CLDGK (Wang & Jiang, 2021b)	1.0	3.4	8.5	13.2	19.3	
ours	1.6	3.3	7.2	11.3	16.7	

But from Top-2 to Top-20 accuracy evaluations, the latest method CL-DKG surpasses our method. The reason might be the fine-grained features strengthen discrimination ability but weaken, to some extent, the generalization ability of the model. However, it can be seen from Table 3, the results of our method are very close to CL-DKG and DGP that are still comparable with the state-of-the-art.

Detailed group comparisons: The most significant contribution of the proposed method is that fine-grained visual features are shared with semantic features in the same knowledge graph. Here, Detail comparisons are aimed to show the effects of such shared visual features through some representative categories of ImageNet. It can be seen from From Table 4, our method significantly surpasses other methods on Top-1 to Top-5 accuracy evaluations in bird, snake, primate and dog categories. Specially, on Top-1 accuracy evaluations, our method even achieves 2-3 times increasing compared with the state-of-the-art. It demonstrates the shared visual features boost the description ability of the knowledge graph that helps to find more unseen classes. And shared visual features also play a greater role than normal semantic features where the model can double the performance with only a small amount of shared visual features added. On Top-10 and Top-20 accuracy evaluations, DGP and our method have the similar performances.

Table 4 Top-k evaluation in detailed group.

Test set	Model	hit@k(%)	
		1	2	5	10	20	
bird	GCNZ (Wang, Ye & Gupta, 2018b)	0.2	0.5	0.9	1.8	3.1	
SGCN (Kampffmeyer et al., 2019b)	2.4	6.2	13.5	21.1	30.6	
DGP (Kampffmeyer et al., 2019b)	2.3	6.0	13.2	20.8	20.2	
ours	4.1	6.9	13.5	20.3	29.6	
snake	GCNZ (Wang, Ye & Gupta, 2018b)	0.2	1.8	4.8	9.9	19.0	
SGCN (Kampffmeyer et al., 2019b)	4.9	10.2	22.8	34.8	49.6	
DGP (Kampffmeyer et al., 2019b)	4.2	9.4	22.8	34.8	49.3	
ours	6.8	12.5	23.0	33.9	47.9	
primate	GCNZ (Wang, Ye & Gupta, 2018b)	0.1	2.2	4.5	6.7	21.7	
SGCN (Kampffmeyer et al., 2019b)	9.7	21.2	44.4	66.8	79.7	
DGP (Kampffmeyer et al., 2019b)	9.6	22.7	49.0	68.6	81.1	
ours	13.1	24.9	49.3	64.5	78.9	
dog	GCNZ (Wang, Ye & Gupta, 2018b)	0.1	2.0	4.4	0.9	14.4	
SGCN (Kampffmeyer et al., 2019b)	6.2	19.4	37.7	46.7	58.0	
DGP (Kampffmeyer et al., 2019b)	6.2	18.6	37.1	46.2	57.1	
ours	17.7	25.3	37.8	46.8	56.7	

Ablation study

As defined in Eqs. (1) and (2), the proposed SVKG mainly consists of Xs, Xv and Xaug. Thus, four combinations, “Xs” only, “Xs+ Xv”, “Xs+ Xaug” and “All”, for ablation study are established. Indeed, “Xs” denotes the traditional knowledge graph while “All” represents the proposed knowledge graph SVKGaug. The experiments are conducted also on the Detailed group (Table 2) to evaluate Top-1 to Top-20 accuracy. The experimental result is presented in Table 5. It is clearly observed that the Top-1 and Top-2 accuracy has significant improvements with the addition of Xs, Xv and Xaug. Specially, in dog category, the Top-1 accuracy has a very steep growth from 6.2% (Xs) to 17.7% (All). The reason is that dog is a well-known category and widely used everywhere. This leads the upstream feature extractor, EfficientDet, to work well and extract more accurate part visual features for SVKG that significantly improves the Top-1 accuracy. A similar phenomenon is also observed in bird and primate categories. The study thus also reveals that the better performance of the upstream feature extractor, the better accuracy the proposed SVKG might achieve. In a short, the study demonstrates the effectiveness of each proposed module in real-world zero-shot learning tasks.

Table 5 The ablation study.

Test set	model	hit@k(%)	
		1	2	5	10	20	
bird	X s	2.3	6.0	13.2	20.8	20.2	
Xs+ Xv	2.9	6.3	13.3	20.8	30.6	
Xs+ Xaug	3.0	6.6	13.8	21.3	31.0	
	All	4.1	6.9	13.5	20.2	29.6	
snake	X s	4.2	9.4	22.8	34.8	49.3	
Xs+ Xv	4.8	10.6	22.4	33.3	46.1	
Xs+ Xaug	6.4	11.9	24.1	35.8	49.7	
	All	6.8	12.5	23.0	33.9	47.9	
primate	X s	9.6	22.7	49.0	68.6	81.1	
Xs+ Xv	10.1	22.9	48.6	64.2	80.7	
Xs+ Xaug	12.0	24.2	51.5	67.4	81.9	
	All	13.1	24.9	49.3	64.5	78.9	
dog	X s	6.2	18.6	37.1	46.2	57.1	
Xs+ Xv	5.3	13.6	34.9	45.2	55.6	
Xs+ Xaug	14.5	24.6	38.9	47.6	57.0	
	All	17.7	25.3	37.8	46.8	56.7	

In the cases from Top-5 to Top-20, the best performance is “ Xs+ Xaug”. The influence of graph augmentation Xaug is weakened after two or three hops, and distant nodes are easy to overfit with the presence of part visual nodes. Therefore, in the long-distance prediction, the overall performance of the model decreases.

Discussions

For further discussion, we conducted an unseen class search compared with DGP on Detailed group. Some representative results are presented in Fig. 5. Obviously, our method obtains better accuracy on unseen class searching than DGP. Interestingly, it can be seen that our method predicts normally more specific class labels than DGP does, such as “plover - sea” (ours - DGP), “Australian blacksnake - lyre snake”, “grivet - old word monkey” and “toy spaniel - toy”. This means more fine-grained information is learned in our model that can be used to predict more specific unseen classes.

Figure 5 The Top-5 unseen classes searching compared with DGP.

The correct label is colored and bold.

Meanwhile, t-SNE was used to visualize knowledge graph features for initial graph, GCNZ graph, DGP graph and SVKG graph on dog category. The orange-colored nodes represent seen classes and the blue-colored nodes represent unseen classes. Figure 6A is the initial graph, where the features are represented by word embedding. It can be found that the graph feature distribution is disorderly and the unseen classes are close to each other, which is not conducive for unseen class predicting. In GCNZ (Fig. 6B) and DGP (Fig. 6C), after cross-modal learning (semantic-to-visual modal), the graph features gradually become clear and show a hierarchical structure. Figure 6D shows the feature distribution of our graph SVKG. Intuitively, SVKG is more structured and order than the graph features of GCNZ and DGP. It can be observed that blue-colored nodes distribute among orange-colored nodes reveals the hidden relationships between seen and unseen classes. This is because the proposed visual features Xv lead the graph representations of SVKG close to real-world dog taxonomy. Unseen class nodes thus are distributed to the most related seen class nodes by the meaning of taxonomy. It obviously alleviates coincidence and proximity during unseen class predicting that improves the performance of zero-shot learning.

Figure 6 t-SNE visualizations for knowledge graph features of initial graph, GCNZ graph, DGP graph and ours SVKG.

Moreover, we also compared feature distribution of SVKG with GCNZ graph feature distribution and real image feature distribution, as shown in Fig. 7. It can be observed that the feature distribution of SVKG is significantly closer to the real image feature distribution than GCNZ. This indicates the feature of SVKG is more close to the real visual feature that is easier to be matched with unseen classes. This also explains why the proposed method achieves the best accuracy on Top-1 and Top-2 unseen class predicting.

Figure 7 The feature distribution of real image feature, GCNZ graph feature, and SVKG feature.

Conclusion

In this article, we propose a semantic-visual shared knowledge graph (SVKG) for zero-shot learning on the real-world dataset without needing pre-defined attributes. It combines semantic (from WordNet and Glove embedding) and visual features (extracted from raw images by EfficientDet) together in the same graph. The visual feature provides detailed information for describing fine-grained semantics that alleviates the “Domain-Shift” problem during the semantic-to-visual transformation of zero-shot learning. A novel multi-modal GCN model is also proposed to learn the graph representations of SVKG. After, the graph representations are further used for downstream zero-shot learning tasks in the experiments.

Experimental results on the real-world dataset demonstrate the effectiveness of our method and illustrate the multi-modal graph guide model generates more discriminative representation. And our method significantly surpasses other methods on Top-1 to Top-5 accuracy evaluations in the bird, snake, primate, and dog categories. Especially, on Top-1 accuracy evaluations, our method even achieves a 2-3 times increase compared with the state-of-the-art.

The important component of zero-shot learning tasks implemented in real-world environments is still how to reasonably use and construct the knowledge graph. In this paper, the SVKG is only storing fine-grained information in parts of visual features. In the future, we will add color, material, shape, and other relations and associated nodes to the SVKG in order to further increase the model’s performance.

Supplemental Information

Supplemental Information 1 Code

Click here for additional data file.

Supplemental Information 2 Supplemental Information 2

Click here for additional data file.

Additional Information and Declarations

Competing Interests

Author Contributions

Data Availability

The authors declare there are no competing interests.

Beibei Yu conceived and designed the experiments, performed the experiments, analyzed the data, performed the computation work, prepared figures and/or tables, authored or reviewed drafts of the article, and approved the final draft.

Cheng Xie conceived and designed the experiments, analyzed the data, prepared figures and/or tables, authored or reviewed drafts of the article, and approved the final draft.

Peng Tang performed the experiments, performed the computation work, prepared figures and/or tables, and approved the final draft.

Bin Li analyzed the data, authored or reviewed drafts of the article, and approved the final draft.

The following information was supplied regarding data availability:

The source code of the proposed method is available in the Supplemental File.

The ImageNet database, taken as a testing benchmark, is available at https://www.image-net.org.

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
