# Peer review of "Semantic-visual shared knowledge graph for zero-shot learning"

_PeerJ Computer Science, doi:10.7717/peerj-cs.1260_

## Round 0.1 · original submission · Major Revisions

Based on reviewers commentary, the authors are requested to make "major revisions".

·

Basic reporting

1. The content is well-written in clear and unambiguous English
2. References are good, but can add few recent papers as well
3. Structure of the articles is good
4. Relevant results are provided

Experimental design

1. Novelty is good
2. As low-dimensional embedding is concentrated with visual semantics, it is technically innovative
3. Methods are described well and with sufficient detail

Validity of the findings

All underlying data have been provided; they are robust, statistically sound, & controlled
Conclusions are well stated

Reviewer 2 ·

Basic reporting

The article should include sufficient introduction and background to demonstrate how the work fits into the broader field of knowledge. Relevant prior literature should be appropriately referenced.

"Domain-aware multi-modality fusion network for generalized zero-shot learning" - https://www.sciencedirect.com/science/article/pii/S0925231222002168
"HSVA: Hierarchical Semantic-Visual Adaptation for Zero-Shot Learning" -https://proceedings.neurips.cc/paper/2021/file/8b0d268963dd0cfb808aac48a549829f-Paper.pdf

Figure 4 is difficult to view. Please enhance the image visibility.

Experimental design

The authors may evaluate the performance of the proposed method in Six traditional benchmark datasets and compare the results obtained in the paper "Domain-aware multi-modality fusion network for generalized zero-shot learning"

Validity of the findings

no comment

Reviewer 3 ·

Basic reporting

Study proposed a semantic-visual shared knowledge graph (SVKG) for zero-shot learning on the real-world dataset without needing pre-defined attributes.

Authors have done good work. I have just few suggestions.

Research questions and their conclusion should be mentioned.
Why this proposed approach is effective please give some statistical analysis.
Conclusion is too general please mention the finding of the study.

Future directions are missing according to study findings

Experimental design

no comment

Validity of the findings

no comment

Additional comments

no comment

Reviewer 4 ·

Basic reporting

See my comments below

Experimental design

See my comments below

Validity of the findings

See my comments below

Additional comments

The authors proposed semantic-visual shared knowledge graph approach for zero shot learning. Problem statement is defined properly. Objectives are clearly stated.

There are major points that need to be addressed as follows:
1. Citation of papers is not proper in the related work section.
2. More surveys on the latest papers are expected for each category in the related work section.
3. Some bad English constructions, grammar mistakes, and misuse of articles: a professional language editing service is strongly recommended to improve the paper’s presentation quality for meeting PeerJ’s high standards.
4. In the problem definition section, a detailed description of the mathematical model is expected.
5. Description of all variables is expected for Eq. No. 2.
6. Rephrasing of sentences is required for line nos. (201-208).
7. Figures 2,3,4,5,6 and 7 are not clear.
8. Examples of seen and unseen labels can be included in the paper.
9. Algorithm of semantic-visual shared knowledge graph can be included in the paper.

---

## Round 0.2 · Minor Revisions

Dear authors,

Revise the paper according to the following comments:
1. Include the explanation for Algorithm 2 as it is important for the readers.
2. Add limitations of your study in the conclusion.
3. Similarly add some numerical results in the Abstract and Conclusion parts.

Reviewer 3 ·

Basic reporting

Authors have done work on my comments and I'm satisfied with the response.

Experimental design

No comments

Validity of the findings

No comments

Additional comments

No comments

---

## Round 0.3 · accepted · Accept

Based on the modifications made by the authors, the paper is now suitable to be accepted.